# Network Construction for Overall Protection and Utilization of Cultural Heritage Space in Dunhuang City, China

**Bin Feng** [1,2,*] **and Yongchi Ma** [3]

1   College of Geography and Environmental Science, Northwest Normal University, Lanzhou 730070, China
2   Key Laboratory of Resource Environment and Sustainable Development of Oasis, Lanzhou 730070, China
3   Gansu Guancheng Planning, Design and Research Co., Ltd., Lanzhou 730070, China
*   Correspondence: fengbin@nwnu.edu.cn; Tel.: +86-13919154300

**Abstract:** An important recent issue in research is the effective protection and rational utilization of cultural heritage. In particular, the regional protection and utilization network of heritage space is the overall requirement for promoting cultural protection and high-quality development of its industry. Using Dunhuang city, Gansu Province, China, as a case study, it is argued here that the cultural heritage space is a living unit that is composed of not only cultural heritage but also its overall environment. By identifying the key historical factors of Dunhuang's regional cultural heritage space, this paper explores the conservation factors and utilization factors. The suitability of the conservation factors and utilization factors is assessed through a two-way index of conservation and utilization. In addition, using a field strength model that considered various factors, the suitability characteristics of conservation and utilization were summarized. It was found that the conservation and utilization space of Dunhuang's cultural heritage had three network features: same level overlap, primary and secondary combination, and significant differentiation. At the same time, these formed an organization network of "patch collage and corridor concatenation" and the network of "mine field pattern and branch extension". From this, the sustainable development of the Dunhuang cultural space network can be realized through the combinations of site protection and ecological protection and environmental utilization and ecological restoration.

**Keywords:** suitability evaluation; space network; field strength model; corridor; Dunhuang

## 1. Network for the Conservation and Utilization of Heritage Space

Currently, China's heritage has been well conserved through careful planning and practices regarding the utilization of large sites. This has played an important role in the conservation, activation, and utilization of the overall environment where cultural heritage is found [1]. However, in the context of increased concern about the conservation and utilization of cultural heritage, the fragmentation and blind utilization of cultural heritage have become widespread problems. On the one hand, China's existing cultural heritage covers part of the world's heritage, cultural relics, intangible cultural heritage, historical and cultural cities, towns and villages, etc., which have diversified protection types and scattered management fields. Cultural heritage is classified into corresponding conservation levels, and the factors are not included in the conservation list such, as regional historical and cultural features with respect to buildings and structures, spatial patterns, etc. [2]. The diversity and complexity of these factors can determine the uncontrollability of the overall protection and utilization. On the other hand, current studies on cultural heritage space principally emphasize large and medium-sized ancient ruins parks, such as the Chang'an Site of the Han Dynasty [3], or they focus on linear cultural heritage sites [4], such as the Great Wall and the Grand Canal. For example, researchers have studied the conservation and utilization of military forts [5]. Alternatively, studies have concentrated on urban or rural historical and cultural settlements as regional geographical units that

span across the administrative boundaries of more than a city. Regional cultural differences are considered, and factors with complex and holistic values are brought together [6]. Thus, most of these studies tend to study a certain theme that covers only part of the entire regional heritage space. More research is needed on the conservation and utilization of the entire regional heritage space.

The Athens Charter on the Restoration of Historic Monuments (also known as the Restoration Charter) in 1931 noted that special consideration should be given to the surrounding environment when approaching cultural relics and historic sites [7]. The subsequent Nairobi Recommendations, Charter of Machu Picchu, Charter of Washington, and Beijing Charter have deepened the connection between cultural heritage and its surrounding environment, highlighting that "historical areas as well as their environment should be deemed as an irreplaceable part of the heritage of all mankind" [8–11]. The Technical Guide for Evaluation of Carrying Capacity of Resources and Environment and Suitability of Land and Space Development (Trial) of the People's Republic of China in June 2019 stated that the cultural conservation space that can be identified covers the legal space for cultural conservation, potential space for cultural conservation, the gathering area of cultural resources, and the contact space for cultural conservation. That is, it is necessary to provide research support for the realization of active protection, effective utilization and overall creation through legal and potential cultural protection space identification and functional structure reorganization. On this basis, we hold that cultural heritage space as a whole is made up of the heritage itself, as well as its overall environment. It covers both the legal and potential cultural space, as well as cultural resource gathering areas and contact spaces. Cultural heritage is composed of human artifacts endowed with historical, scientific, artistic, and cultural value that are shaped through human interaction. This includes cultural relics, ancient buildings, and sites, as well as their production and living environment. The overall environment is a region shaped by natural, cultural, and artificial environments that depend on cultural heritage, covering the historical environment and the cultural landscape. This includes visible material forms and the natural and artificial backgrounds associated with these material forms, as well as the background of the historical environment linked with the society, economy, and culture.

Since the Second World War, all countries have begun to explore network components such as conservation zones and cultural routes. Regarding conserved zones, the Nairobi Recommendations (1972) proposed the concept of "historical areas" [8]. Since then, historical areas, historical features, and utilized areas of landscape have been established in Japan, and buildings, cities, and landscapes have been conserved in France, alongside conserved zones in the UK and zoning in the US [12]. Thus, these countries are principally developing conserved zones and comprehensive zones for heritage and the environment, and even regional conserved zones that cover buildings, cities, and landscapes. With regard to cultural routes, the ICOMOS Charter of Cultural Routes was officially adopted by the International Council on Monuments and Sites in 2008. It made provisions on the definition and features of cultural routes [13]. The sustainable conservation of cultural routes and their environment is beneficial to cultural exchange and dialogue [14,15], as well as to the development of heritage in urban and rural areas and the promotion of social cohesion [16].

Currently, research on networks for the protection and utilization of cultural heritage has become invaluable. Plans for the construction of an ideal heritage space system are based on the international consensus regarding the protection and utilization of cultural heritage, as well as practical domestic needs. For instance, the Cultural Heritage Conservation Law revised in Japan in 2018 supplemented the identification system of the integral conservation and utilization plan of cultural heritage, and moved it down to the regional level to achieve integrated conservation and utilization. This included drawing up an outline for protection and utilization, identifying regional planning, designating support groups, and determining an implementation plan that aimed to strike a balance between protection and utilization. Such an approach will not only contribute to social and economic development, but also help in achieving cultural revitalization and enriching

residents' lives. In addition, researchers have sought to cover specified and unspecified heritage objects and laid an integral foundation for heritage studies [17]. Another example is Australia, where the protection and utilization of industrial heritage is both systematic and comprehensive [18]. Thus, research on cultural heritage needs to proceed from the overall perspective of heritage and its environment and consider its conservation and utilization. Scholars in China have launched research on the concept of a heritage corridor network [19], the construction of a space network for the conservation of settlement-type cultural heritage [20], spatio-temporal distribution of tourism flows, network analysis of traditional villages [21], and the construction of regional eco-cultural network [22], but there is relatively little research on network spatial patterns that consider both conservation and utilization.

In this context, it is particularly significant to study the overall spatial network relationship from a regional point of view in combination with the existing conservation system of hierarchical classification of various countries. Here, it is requisite to launch investigations, analysis, and development of overall spaces with emphasis on particular regional cultures. On the one hand, the use of a dual evaluation method of conservation and utilization of cultural heritage spaces is needed to contribute to the scientific and reasonable development of such spaces. On the other hand, it is further necessary to place cultural heritage in the spatial structure of the historical environment, and to comprehensively study the material and intangible cultural heritage in regions while taking into account the continuity and correlation of culture. Doing so involves covering the landscape patterns of the region, the morphological structure of urban spaces, and the linear correlation of urban and rural settlements to construct a spatial network of conservation and utilization from a regional perspective [23].

## 2. Research Area and Research Methods

### 2.1. Overview of the Research Area

Dunhuang City, Gansu Province, China, was chosen for the present study. The cultural space of Dunhuang is basically presented as historical relics and a memento. Dunhuang not only is a place where Chinese civilization blends with the cultures of Central Asia and other countries in grotto art, the city is also a famous historical and cultural city in China, the westernmost section of the Great Wall National Cultural Park. Under the Belt and Road Initiative, it will develop into a cultural exchange center along the Silk Road, the road network from Chang'an to Tianshan, a world cultural heritage site. This will involve building a network of overall conservation and utilization of the cultural heritage space, so that its culture can be converged, regenerated, and exported.

Dunhuang city is situated at the western end of the Hexi Corridor in Gansu Province, against the backdrop of the Qilian Mountains, the Shule River basin, and Dang River tributaries. A number of ecological functional areas of natural or artificial oases of different scales take shape thanks to the nourishment of the dendritic river system, including the Dunhuang Oasis and the Yangguan Oasis. These oasis environments interlace with the Gobi desert around them, forming an interrelated landscape environment and geomorphic pattern of complementarity and coexistence. In such environmental conditions, there are more than 240 tangible cultural heritage sites. There are also more than 20 intangible cultural heritage objects, including immovable cultural relics, famous national historical, and cultural cities, which have been suggested for conservation in the world, national key provinces, cities, counties, and cultural-relic surveys (see Figure 1) [24]. Depending on the types of cultural heritage, they can be summarized as pertaining to military defense, the postal system, historical and cultural cities and towns, the ancient Silk Road, grotto art (and religion), and intangible cultural heritage.

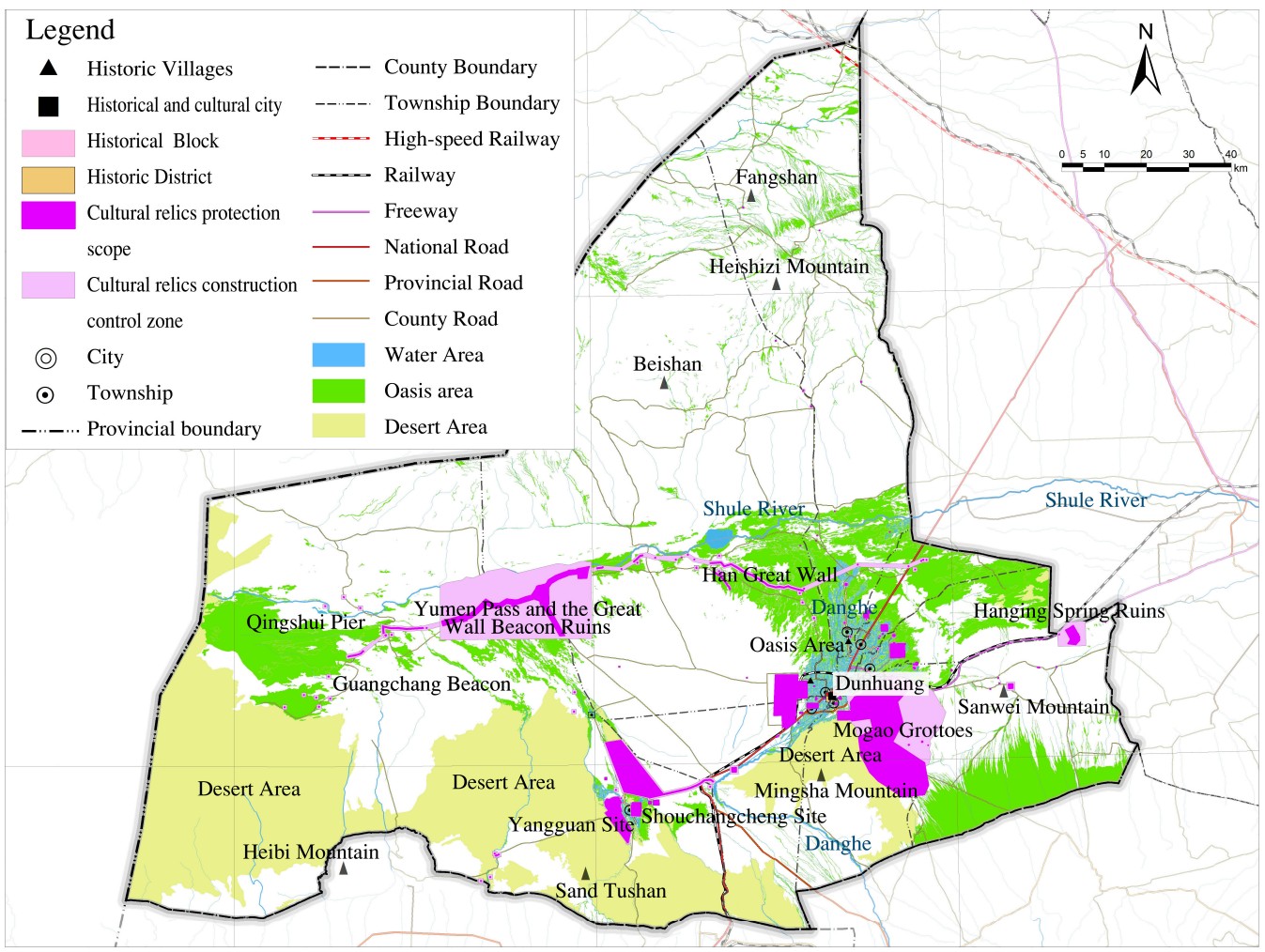

**Figure 1.** Distribution of Dunhuang cultural heritage and its overall environmental factors. Source: drawn by the author (2022).

*2.2. Data Sources*

The data of tangible cultural heritage studied in this paper come from the vector data of the conservation scope of cultural heritage in Dunhuang's land and spatial planning for the year 2020. Data on intangible cultural heritage and its transmission and seminar are derived from the Conservation Planning of Famous Historical and Cultural Cities in Dunhuang [24]. According to the data collation, the conservation scope and construction control area of national key provincial, municipal, and county-level cultural relic-conservation units in Dunhuang comprise 498.47 km$^2$, 240.23 km$^2$, and 39.31 km$^2$, respectively, totaling 778.01 km$^2$ and making up about 2.5% of the administrative area of Dunhuang City. Thus, its legal cultural space takes up a relatively large part of the city. Meanwhile, topographic data come from the geospatial data cloud. Data regarding the river water system, land use attributes, and other data are derived from the third land survey data. Data on the ancient Silk Road in Dunhuang come from "Roads and Traffic in Dunhuang" in Dunhuang's History and Geography (Binglin Zheng) [25]. Data on military defense facilities of the Great Wall are derived from the Survey and Research of the Hansai in the West of Yellow River (Wu Rengxiang) [26]. Data on the Mogao Grottoes and other ancient cave temples are derived from the Master Plan for the Conservation of Mogao Grottoes in Dunhuang (2006–2025) and survey data on Dunhuang cultural relics. Because of the different data sources, the third land survey data are used as the basic map in the research to ensure the consistency of the data.

*2.3. Research Methods and Technical Process*

2.3.1. Quantitative Methods and Process of Indexes

The factors of the cultural heritage space in Dunhuang city can be classified into many types, and it is requisite to evaluate the suitability of each type. The overall distribution rules and characteristics of different cultural spaces can be identified through kernel density analysis, buffer analysis, etc., which have been applied in many studies and practices [27]. So, the multiple evaluation indexes of a single factor are scored by experts, the weights are analyzed by AHP, and the data space is analyzed by kernel density analysis, buffer analysis, neighborhood analysis, and other methods. Some indexes can be used to measure the distribution density (aggregation degree) and distribution pattern of various factors by means of kernel density analysis. The larger the kernel density value, the greater the distribution density of factors and the more prominent the distribution pattern. The formula for the kernel density analysis is as follows:

$$F_n(x) = \frac{1}{nh^2\pi} \sum_{i=1}^{n} k \left[ 1 - \frac{(X - X_i)^2 + (Y - Y_i)^2}{h^2} \right]^2 \tag{1}$$

where $F_n(x)$ is the kernel density value, and $h$ is the evaluation parameter value, indicating the bandwidth; $n$ is the number of points in the analysis area, and $k$ is the kernel function; $(X - X_i)^2 + (Y - Y_i)^2$ indicates the distance from point $(X_i, Y_i)$ to $(X, Y)$.

The neighborhood of another part of the indexes can be determined in certain spatial objects or groups by way of buffer analysis. For the specified object $A$, its buffer can be defined as follows:

$$P = \{x \mid d(x,A) \leq r\} \tag{2}$$

The spatial field strength model in geography is mostly applied to urban space, cultural space, regional analysis, and other aspects. For example, the spatial recognition and pattern evolution of cultural memory in Gansu province, China [27,28]. Thus, with GIS technology, the spatial field strength model can be used to evaluate the factors of the cultural heritage space. Under the ideal condition of no interference factors, physical field strength theory is used to determine the conservation or utilization of the factors of cultural heritage space as the distribution of the "magnetic field", and its size is based on the size of the "field strength". The calculation is Equation (3). The field strength model is chiefly influenced by two variables: the multi-factor cost-weighted value of conservation or utilization of cultural heritage space, where the larger the value, the stronger the field strength, as given in Equation (4), and the spatial distance from its corresponding cultural heritage, where the smaller the distance, the greater the field strength. The formula is as follows:

$$E_{ij}^p = \frac{Z_p}{(D_{ij}^p)^f} (p = 1, 2, \ldots n) \tag{3}$$

$$Z_p = \sum_{x=1}^{n} W_x A_x \tag{4}$$

where $(i, j)$ represents the position of a certain factor in space, $E_{ij}^p$ is the field strength of factor $p$ at $(i, j)$, $Z_p$ is the weighted value of factor $p$, $D_{ij}^p$ is the spatial distance of factor $p$ at $(i, j)$, and $f$ is the friction coefficient of distance. Usually, the standard value is 2.0, where $W_x$ is the weighted value of factor $x$, $A_x$ is the evaluation value of the $x$th factor, and $n$ is the total number of factors.

Therefore, in the whole research process, first of all, each cultural heritage spatial element is evaluated by the conservation index, or by using the utilization index. The expert scoring method can be used to obtain the protection evaluation score of each cultural heritage spatial element or utilization evaluation score. Then, by using the field strength

model and integrating the evaluation scores of various cultural heritage elements, the overall protection evaluation and grading of Dunhuang city can be obtained. (see Figure 2)

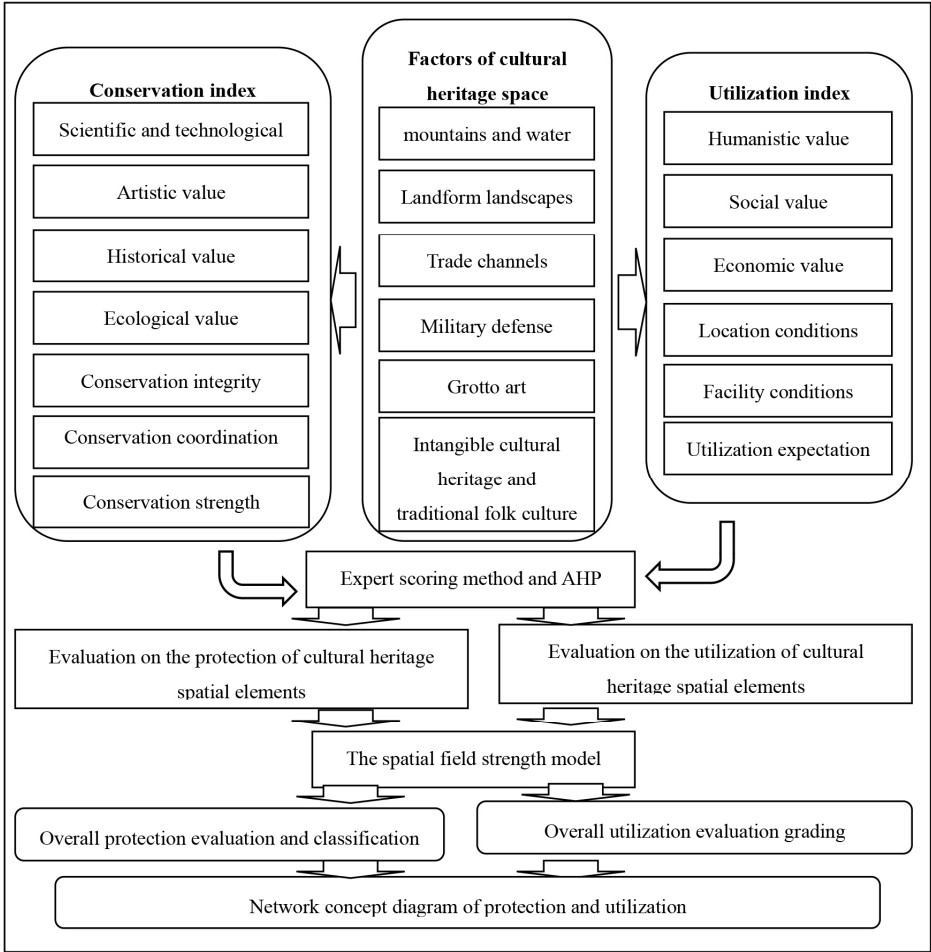

**Figure 2.** Method and technical route. Source: drawn by the author (2022).

2.3.2. Index System for the Suitability Evaluation

In the revision of the Operation Guide in 2011, it was clearly stated that "the most significant thing for the sustainability of heritage is to conserve its outstanding universal value". Hence, the conservation and utilization of cultural heritage cannot be separated from the research orientation of the universal value [29,30]. Based on the evaluations of the protection of cultural relics in the Burra Charter [31], from the perspective of public ideas about cultural heritage and its overall value to the environment, it is deemed that the cultural heritage space in the public's mind as a whole is not only concerned with scientific, technological, artistic, and historical value, but also ecological, humanistic, social, and economic value at a deeper level [30,32–34]. We propose a two-way index for a suitability evaluation of the conservation and utilization of the spatial factors of cultural heritage. The proposed index covers the value of both conservation and utilization. On the one hand, conservation principally gives consideration to two aspects: conservation value and conservation basis. Therein, conservation value principally indicates whether or not the value is scientific, technological, historical, artistic, and ecological, and to what extent. These indexes are derived from part of the value composition of the cultural heritage space as a whole. The conservation foundation is chiefly used to estimate the environmental integrity, coordination, and conservation efforts of the situation and mutual relationship of conservation work [35]. Hence, the necessity and feasibility of conservation are expounded herein. On the other hand, utilization mainly gives consideration to two aspects, i.e., utilization value and utilization foundation, wherein utilization value chiefly represents

the extent of humanistic, social, or economic value. These indexes are also derived from part of the value composition of the cultural heritage space as a whole. The utilization foundation is principally used to examine the development of locations, facility conditions, utilization expectations, etc., which together embody the natural and artificial construction foundations and the shared cultural memory.

The importance of a "conservation factor" and the feasibility of a "utilization factor" are determined through the suitability evaluation of the "conservation index" and "utilization index", respectively, of the spatial components of cultural heritage. The index values are all between 0 and 1 (see Table 1). Apart from that, in determining the scale and proportion of conservation factors and utilization factors, the essential features and dominant position of the cultural heritage space should be adhered to. The natural constraints, social needs, and market dynamics of cultural resource utilization should be fully connected.

**Table 1.** Main indexes and connotations of the suitability evaluation of the factors of the cultural heritage space source. Written by the author (2022).

| Conservation Representation | Conservation Index | Interpretation of Connotation | Utilization Representation | Utilization Index | Interpretation of Connotation |
|---|---|---|---|---|---|
| Conservation value | Scientific and technological value (P1) | It embodies people's capability and aesthetic concept of understanding nature and transforming the natural environment, as well as the advanced nature of manufacturing technology. | Utilization value | Humanistic value (U1) | It is beneficial for the facilitation of cultural conservation and spatial memory, enhancing local cultural self-confidence, excavating local culture and national spirit, and promoting cultural communication and interaction. |
| | Artistic value (P2) | It embodies the social system, social production, and social living environment of all ages and nationalities in history. | | Social value (U2) | It is beneficial for the facilitation of people's communication and health, enhancing production and the living environment, promoting dialogue between history and reality, and optimizing the regional labor force. |
| | Historical value (P3) | The environment influenced by human beings may be in possession of the value of arts and crafts in numerous aspects, such as shape, decoration, and color. | | Economic value (U3) | It is beneficial for the imporevement of the local economic structure, facilitating the development of cultural tourism, advancing the revitalization of rural culture, and achieving the integration of the three industries. |
| | Ecological value (P4) | It can help maintain ecological security and stability, improve the landscape, promote ecological and organic restoration, and encourage the construction of ecological civilization. | | | |

**Table 1.** *Cont.*

| Conservation Representation | Conservation Index | Interpretation of Connotation | Utilization Representation | Utilization Index | Interpretation of Connotation |
|---|---|---|---|---|---|
| Conservation foundation | Conservation integrity (P5) | It is complete in the overall pattern, and comprehensive in the physical composition. | Utilization foundation | Location conditions (U4) | It enjoys a good comparative advantage of natural and human resources, and is closely linked with other spaces of utilization factors. |
| | Conservation coordination (P6) | It is mutually associated in terms of system composition, and orderly in the relationship between heritage and the environment. | | Facility conditions (U5) | It lays a supporting foundation for public facilities and infrastructure, offering certain conditions for utilization. |
| | Conservation strength (P7) | The conservation and research work is in depth, and stability restoration work achieves fruitful results. | | Utilization expectation (U6) | It arouses more attention and attraction, and the public has high expectations for its utilization. |

## 3. Refinement and Suitability Evaluation of Historical Information of Conservation and Utilization Factors of the Cultural Heritage Space

The basis of this research is to set up a complete system of the factors of the cultural heritage space. To sort out Dunhuang's cultural heritage and its environmental composition and changes, we should, at the city level, emphasize natural space factors such as mountains, water systems, and the landscape, and artificial space factors such as trade channels, military defense communication, grotto art, intangible cultural heritage, and traditional folk cultural sites. Therein, trade channel covers the county(state)–town–township system, the ancient Silk Road, etc., and military defense communication covers military defense systems, postal systems, etc. These natural and artificial factors are interrelated and integrated at the global level. On the one hand, artificial factors such as defense systems, post-traffic systems, ancient trade roads, and grotto art take shape under the action of natural factors. On the other hand, artificial factors influence natural factors, principally manifested by changing the local water systems, oases, and other natural environment patterns to some extent.

The factors of cultural heritage space are derived from the refinement of information in different periods. Japan's "metabolic city" theory emphasizes the historical process of urban development and change, the key vital factors in different historical stages, and the development process for conservation and re-utilization [36]. Since the Han Dynasty, Dunhuang has gone through the historical period of the Central Plains Dynasty. Thereafter, along with the changes in Dunhuang's cultural heritage, such as its construction, existence, abandonment, and decline [25], its time can be classified into the construction period (Han Dynasty), existence period (Wei, Jin, Southern, and Northern Dynasties, Sui Dynasty, the early Tang Dynasty, the Tubo occupation period, and the period of the Gui-yi-jun Regime), abandonment period (Xixia, Yuan, and early Ming Dynasty), declining period (the middle and late Ming Dynasty and the period since the Qing Dynasty, including the People's Republic of China and modern times), and the conservation and utilization period (since reform and opening up). The historical spatial factors of Dunhuang city since the Han Dynasty are sorted out in stages with varying emphasis. Therein, the construction period covers the generation process of cultural heritage, the existence period embodies the survival or continuation of cultural heritage, the abandonment period bears the loss or abandonment of cultural heritage, the decline period witnesses the decline and extinction of

cultural heritage, and the conservation and regeneration period involves the conservation, inheritance, and utilization of cultural heritage.

*3.1. Refinement of the Historical Information of Various Factors in Different Periods*

Regarding mountains and water systems, many low mountains in front of the Qilian Mountains are integrated into grottoes, folk customs, and a variety of other historical and cultural factors that embody the culture in mountainous regions, such as the Mingsha Mountains and Sanwei Mountains, which are the most typical. River wetlands have changed to a certain extent in local areas, from decreased and interrupted ancient water systems to the reappearance of surface water, and a local oasis environment was gradually formed by intermittent rivers or sudden springs. These reflect the features of changes to historical environmental factors due to the distribution of water [23,35]. In recent years, thanks to the improvement of the ecological environment, some historical environmental phenomena such as seasonal rivers and local sudden springs have gradually recovered.

Regarding landscape factors, due to the changes of rivers, the geomorphic landscape has also changed accordingly, taking on the features of the alternating succession of wetlands, oases, and deserts. For instance, the Dunhuang Oasis is relatively stable, although desertification exists in the eastern and western marginal areas. Even so, some oases in the north have increased, reflecting the inheritance and development of the oasis environment [37]. Regarding the factors of trade routes, with the change of the smoothness and obstruction changes to the Silk Road, Guasha Road, Nandao Road, Dahai Road (also known as Liuzhong Road), Daqi Road, Shuogandao Road, and Nanshan Road, with Dunhuang as their nodes, have gone through ups and downs over time, resulting in variation in trade [25]. Some of these ancient roads are close to modern roads, characterized by good cultural plasticity. They are the main body of the Dunhuang World Cultural Heritage Routes.

In terms of the factors of military defense communication, due to the protection of the frontier fortress and the management of the western regions, the postal system gradually flourished in the Han Dynasty. The military defense system of the Great Wall of the Han Dynasty covered the beacon line of the Great Wall in the north, as well as the surrounding block wall and houses in the south. (see Figure 3) The post-traffic system covered the existing sites of Xuanquanzhi and the environment of Yulizhi, laying the foundation for the cross-domain network pattern of Dunhuang [38]. In other periods, military defense facilities were partially supplemented. With regard to the artistic factors of grottoes, the northern and southern dynasties started excavating the grottoes, which continued and added in other periods, forming a Dunhuang grotto group consisting of the Mogao Grottoes, West Thousand Buddha Cave, Dongshuigou Grottoes, and Nanhu Diandong Grottoes (see Figure 4). Intangible cultural heritage and folk culture have experienced long-term multi-ethnic cultural integration, accumulation, and precipitation, and formed unique local folk cultural activities, such as the Quzi Opera (national intangible cultural heritage), including many seminars such as the Suzhoumiao Village in Suzhou Town and Chenjia Culture Courtyard in Yueyaquan Town. These seminars are distributed in the towns and villages where Dunhuang and the Yangguan Oasis are situated [24].

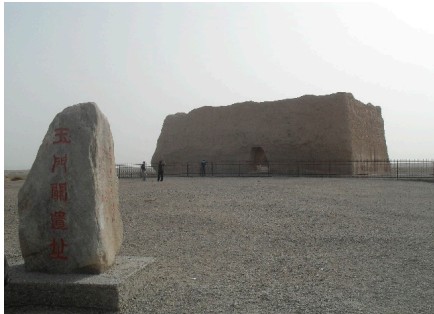

**Figure 3.** Yumenguan Site, one of the elements of military defense communication. Source: authors' photographs (2021).

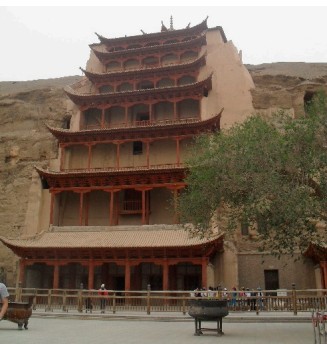

**Figure 4.** Mogao Grottoes, one of the artistic factors of grottoes. Source: authors' photographs (2022).

*3.2. Suitability Evaluation of Conservation and Utilization of Various Factors*

By refining the key information of the factors of the cultural heritage space in Dunhuang City, and by combining the actual conservation, preservation, management, and utilization of these factors, we can judge the importance of conservation and the feasibility of the utilization of each factor of the cultural heritage space. From the analysis of the superposition of cultural heritage, mountain and water systems, and landform landscape, the conservation value and foundation of the front area of the Qilian Mountains and the middle reaches of the Danghe River, where cultural heritage is concentrated, are high. Meanwhile, owing to the existence of water, the utilization value and foundation along the Dang River and Shule River are more prominent. From the analysis of the composition of the cultural heritage of trade routes, military defense communications, and historical identification information, we found that existing sites, such as the ancient Silk Road, the Great Wall, and the Xuanquanzhi, are provided with high conservation value and foundation. In contrast, ancient roads, the Great Wall, and the post-traffic stations that have died out but appear in historical records, poems, and cultural notes are of high utilization value. Particularly in the construction of the Great Wall and the National Cultural Park, the Great Wall defense system and supporting facilities for transportation and post-traffic stations are of high utilization expectations. According to the analysis of the conserved zones of the Dunhuang Grottoes, the conservation scope has higher conservation value and foundation. By contrast, the peripheral construction control area has more value and foundation for development and utilization. Regarding intangible cultural heritage and traditional folk culture, the oasis area bears the chief functions of inheriting, spreading, and displaying these cultures. Its conservation value and foundation are high. Meanwhile, there are high expectations for a cultural scene reproduction in the oasis area of human settlements in Dunhuang. On this basis, we evaluate the protection or utilization index of each element by expert scoring in turn, and we determine the weight by AHP to calculate the evaluation score of the factor. The analysis results of the data space of each factor can be derived through the publicity of 1 and 2, which embodies the evaluation level of protection and the utilization of each factor (see Figure 5, Table 2).

Next, we calculated the results of the suitability evaluation of the conservation and utilization of the factors of the Dunhuang region cultural heritage space using the field strength model (publicity 3 and 4; see Figures 6 and 7). Conservation and utilization can be classified into the first-class corridor or partition, which is significant for conservation or utilization, and the second-class corridor or partition, which is generally significant for conservation or utilization. From this, the conservation space of cultural heritage is the representative symbol of the historical space in different periods. It is not connected in space and takes on a fragmented status, with a small part of it showing local partitions or corridors, whereas most of it is in tiny scattered patches. In combination with the evaluation process of sub-factors, the spatial differentiation features of Dunhuang's cultural factors are obvious. These patches themselves usually carry obvious types of themes, such as the Great Wall military defense and its environment, grottoes and their environment, etc., thereby forming the characteristics of the organization of the cultural heritage space. Moreover,

the utilization space of cultural heritage is characterized by a certain continuity. This can supplement and expand the overall historical network structure, taking on a superposition phenomenon with the conserved space, or it can be manifested by a further extension on the basis of the conserved space, thus forming the cross-domain linear features of the ancient Silk Road, the Great Wall, and other factors. It can be seen from this that only with the utilization space can the formation of the pattern of linear heritage settlements be promoted. The tourism service industry, leisure agriculture and forestry, ecological construction, etc., can be appropriately developed according to the local conditions.

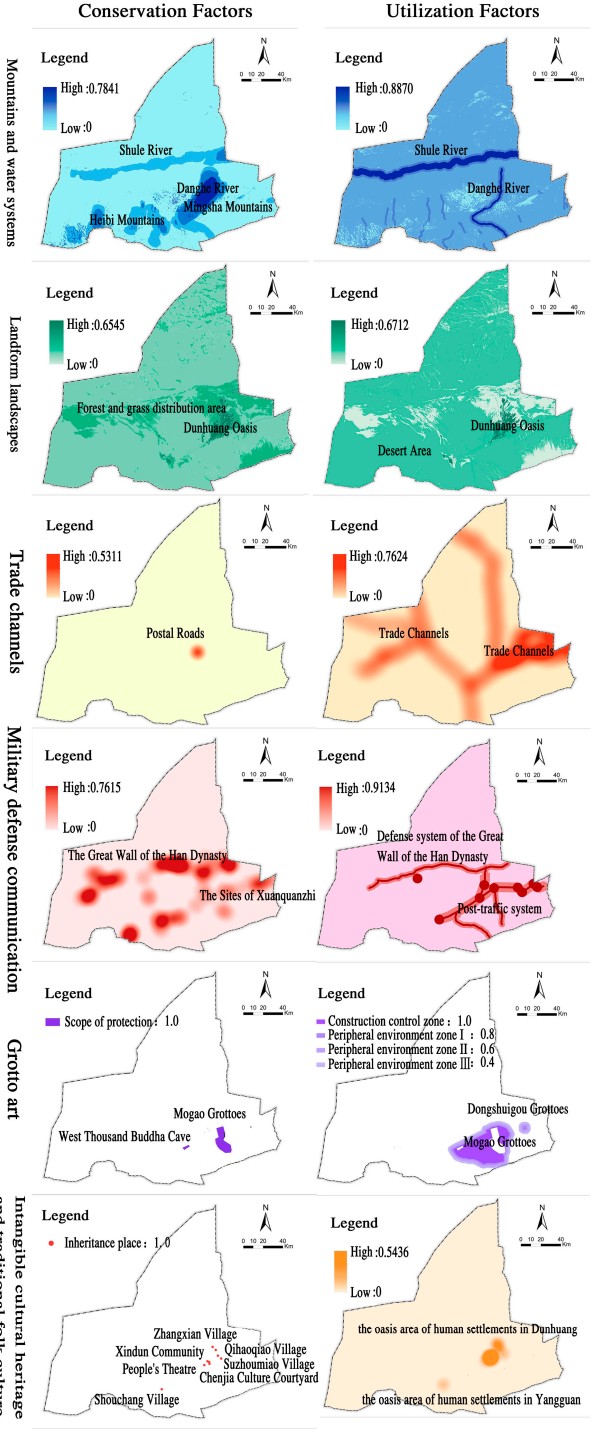

**Figure 5.** Evaluation of the factors of conservation and utilization of cultural heritage space. Source: drawn by the author (2022).

**Table 2.** Evaluation process explanation of the factors of conservation and utilization of cultural heritage space. Source: written by the author (2022).

| Factors of Cultural Heritage Space | Conservation Evaluation | | | Utilization Evaluation | | |
|---|---|---|---|---|---|---|
| | Analytical Method | Assignment (1~0) | Weight | Analytical Method | Assignment (1~0) | Weight |
| Mountains and water | Kernel density analysis | The protection value and foundation of the front area of Qilian Mountains and the middle reaches of Danghe River scored high | 0.0991 | Kernel density analysis | The utilization value and foundation scores of Danghe River and Shule River are high | 0.0898 |
| Landform landscapes | Neighborhood analysis | The score of protection value in oasis area is high | 0.1213 | Neighborhood analysis | The score of basic utilization in oasis area is higher, followed by Gobi area | 0.2243 |
| Trade channels | Buffer analysis | The protection value score of the existing ancient road remains is high | 0.1634 | Buffer analysis | The utilization value and basic scores along the historic road are high | 0.1767 |
| Military defense | Kernel density analysis | The protection value and foundation score of the Great Wall site is good | 0.3105 | Buffer analysis | The utilization value and basic score of the Great Wall National Cultural Park and the post road are good | 0.2565 |
| Grotto art | Neighborhood analysis | The core protection area of the grottoes has a high score of protection value | 0.1945 | Buffer analysis | Grotto construction control zone and surrounding utilization base score is good | 0.1652 |
| Intangible cultural heritage and traditional folk culture | Point element analysis | The protection and inheritance value and foundation score of the institute is high | 0.1112 | Buffer analysis | The value of cultural inheritance and development in the residential oasis is great | 0.0875 |

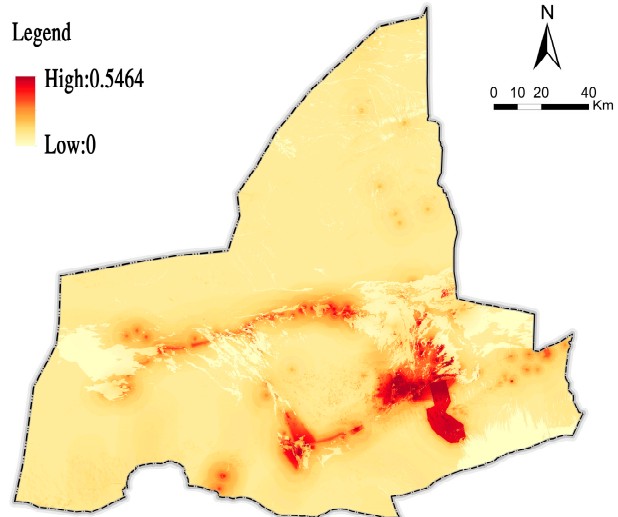

**Figure 6.** Overall protection evaluation and classification. Source: drawn by the author (2022).

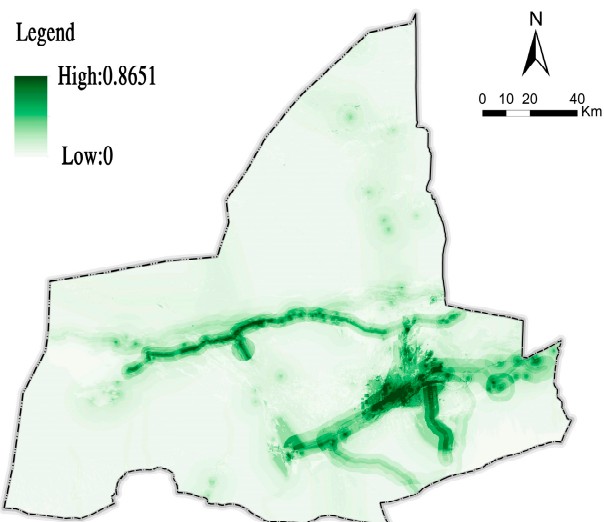

**Figure 7.** Overall utilization evaluation grading. Source: drawn by the author (2022).

## 4. Network Construction of Conservation and Utilization of the Cultural Heritage Space

Based on the suitability evaluation of the conservation and utilization above, it can be concluded that the space for the conservation and utilization of cultural heritage in Dunhuang is neither a simple corridor nor a patch or partition of unconnected links. Rather, it is a spatial network shaped by the combination of closely linked corridors and patches for conservation and utilization.

We hold that there are three features of this network in Dunhuang: overlapping at the same level, the combination of primary and secondary spaces, and significant differentiation. First, the overlapping at the same level involves a complex network of the same level of conservation space and utilization space with overlapping partitions or corridors. This is characterized by "One level and the same level" (such as the Dunhuang historic district) or "The second level is the same as the second level" (such as the contact space between Dunhuang district and Yangguan). Here, the conflict between conservation and utilization is obvious. We cannot simply consider conservation or simply achieve utilization. The proportion of conservation and utilization varies with the concrete types of cultural heritage. Second, the combination of primary and secondary spaces covers the conservation and utilization network with conservation as the main body (such as the sites of Xuanquanzhi and its surrounding spaces), and the conservation and utilization network with utilization as the main body (such as the contact space from Yangguan to the sites of Yumenguan). This is characterized by the conservation or utilization of the main factors, while the utilization or conservation of other corresponding factors is relatively secondary. Third, significant differentiation is obvious in the conservation or utilization of the area. This is reflected by the dominant network of conservation or utilization. For example, the environmental corridor along the central Duwei section of the Great Wall of the Han Dynasty, where the Dang River and Shule River meet, is mainly subject to utilization, taking into account the conservation factors.

Combined with the network types above, we can also construct a diagram of the network, which shows the integral conservation and utilization of the cultural heritage space in Dunhuang city (see Figure 8). On the one hand, the network is a collage of patches and a series of connections. This collage of patches principally refers to the mutual collage of primary and secondary protective patches, while the possibility of a collage of utility patches is small. These collage patches are closely attached to the conservation and utilization corridors. The series of connections refers to the conservation and utilization corridors, conservation corridors, and utilization corridors and their connections to each other. The main corridors in Dunhuang are not made up of either conservation or utilization, but shaped by block combinations. For example, Yumen Duwei along the Great Wall of the

Han Dynasty is mainly a first-class conservation and utilization corridor. Central Duwei is connected in series on its east side, and it is mainly a first-class utilization corridor. These embody the accumulation of the cultural space of the Great Wall of the Han Dynasty, as well as the natural constraints and utilization of oases, water systems, etc. They contribute to the formation of the cultural exhibition belt of the Great Wall of the Han Dynasty in series, and to the construction of the national cultural park system of the Great Wall of the Han Dynasty [39]. On the other hand, a network of grid patterns and branched extensions takes shape, in which grid patterns are horizontally connected by the corridor of the Great Wall of the Han Dynasty, the National Cultural Park in the north, and the postal road corridor of Guasha Road and post-traffic system in the south. It is vertically connected by the Yangguan Duwei Corridor and the cultural ecological corridor of the Danghe River in the southern road of the western regions (from Yangguan to Yumenguan). This forms a "Two horizontal and two vertical"-shaped network. Moreover, block-like patches are shaped at the local node of "Two horizontal and two vertical", which is inseparable from the corridor. A patch that is "block"-shaped forms, owing to the partition of the corridor. Branched extensions are principally distributed in the low mountainous areas in the south, with a certain extension along the river valley. For instance, the branched corridor of the Mogao Grottoes extends to the upper reaches of the Daquan River for exhibition and sightseeing. This is helpful to the integral conservation, zonal display, and node utilization of this cultural heritage.

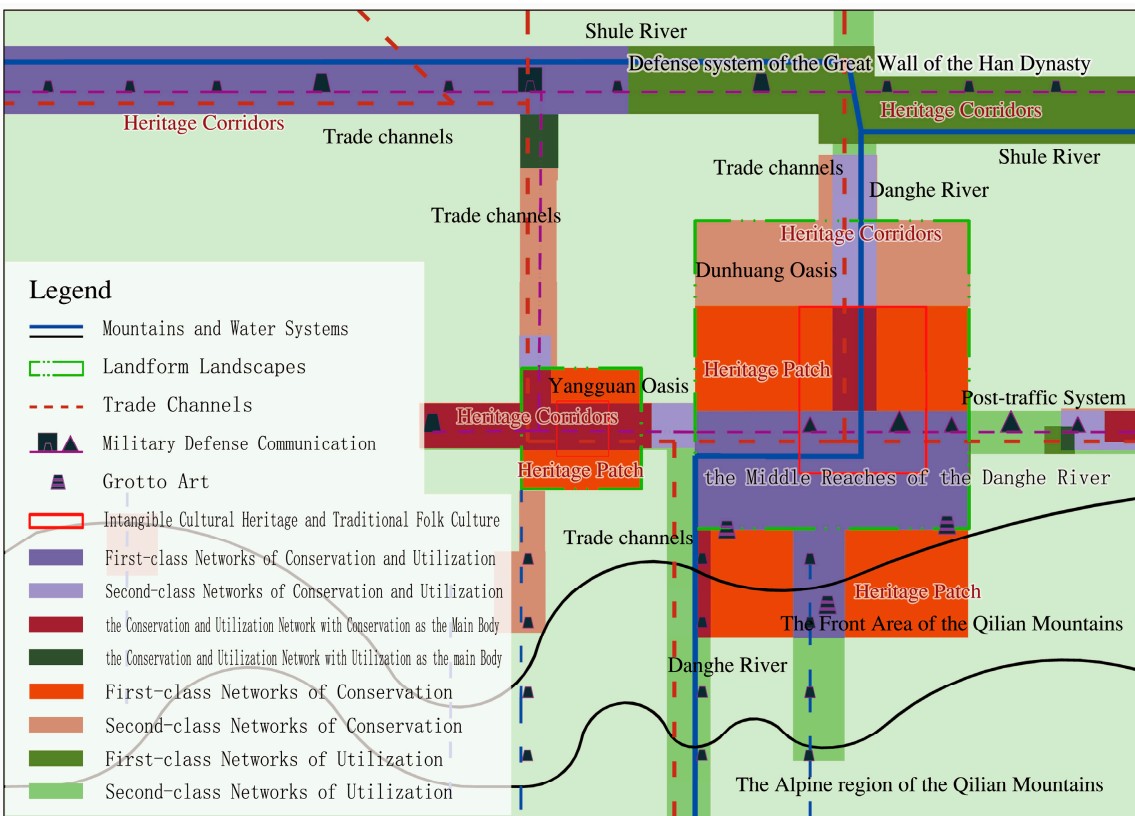

**Figure 8.** Network concept diagram of conservation and utilization in the cultural heritage space of Dunhuang city. Source: drawn by the author (2022).

The grid organization above shows that the conservation patches of the Dunhuang cultural heritage are collage-like, rather than having a simple concentric structure. That is, the importance of heritage conservation decreases from the center to the periphery. Furthermore, heritage corridors are shaped by connecting conservation and utilization in series, instead of the central line of the corridor being the conservation space and the periphery being the utilization space. Hence, only by constructing a reasonable utilization

space can the conservation spaces be fully linked. Furthermore, patches and corridors are nested within each other, rather than separated from each other. These features are subject to the flow direction of water systems and water distribution, and are determined by the constraints of the natural environment pattern of mountains, oases, and the desert. The network embodies a matrix of oasis landscape ecology. These features reproduce the composition and essence of regional culture, as well as the cultural lifeblood of the ancient Silk Road system and the Great Wall military defense system as a whole. They demonstrate the coupling of cultural and social ideology with the natural environment. Hence, the conservation and utilization of Dunhuang's cultural heritage space should enhance the organic management of landscapes, forests, lakes, and grassland. The space should include historical and cultural landscape restoration and the sustainable development of Dunhuang's cultural space. This can be achieved by combining site conservation with ecological conservation, environmental utilization, and ecological restoration.

## 5. Conclusions and Discussion

By paving the way for the research on networks of integral conservation and utilization of cultural heritage spaces, this paper considered Dunhuang as a case study and evaluated the suitability of the conservation and utilization of its constituent factors. We developed a two-way index of conservation and utilization for this evaluation. The index can be used to assess the conservation and utilization of cultural heritage, given the differences and connections in the respective aims of conservation and utilization. In this process, the field strength model illustrates the suitability features of conservation and utilization. We found that the conservation and utilization space of Dunhuang's cultural heritage is presented in three network characteristics: overlapping at the same level, the combination of primary and secondary spaces, and significant differentiation. Dunhuang formed an organization network of "patch collage and corridor concatenation" and the network formed a "mine field pattern and branch extension". The cultural heritage space is a product of its natural and humanistic environments, and these environments are the conditions for the conservation and utilization of the cultural heritage space.

In the overall network environment of land space, the relationship between three districts and three lines, the patterns of land space conservation and utilization, and the conservation and utilization of cultural heritage need to be further connected. With strict constraints on ecological space, production space, and living space, a network of conservation and utilization for cultural heritage spaces can be helpful in guiding the construction of regional cultural functions and for selecting cultural products. The results of our study can guide the conservation, activation, and utilization of cultural resources from the overall level of the city, and help facilitate the management of cultural heritage spaces.

With the development of social networking at this stage, the traditional and simplified heritage management model is facing unprecedented challenges. In this context, the space conservation and utilization network of cultural heritage should promote the development of regional cultural heritage management and control. At the regional level, it is necessary to ensure the bottom line of cultural safety, life safety, conservation safety and ecological safety, highlight the overall spatial pattern and basic cultural development characteristics of culture, highlight the advantages and characteristics of cultural heritage, promote the inheritance of high-quality cultural heritage, improve social quality, and develop economic capacity. By coordinating and empowering regional cultural resources, the cultural landscape pattern is presented in the form of cultural space (i.e., combination of cultural resources and cultural ecology and its spatial form) in the historical environment gathering area and corridor of cultural heritage, reflecting the overall spatial control. It is worth discussing that on the basis of the above regional level, a detailed level guidance is also needed. Based on the composition of cultural heritage inside and outside the area, the cultural space unit structure of historical environmental conservation and utilization should be sorted out to reflect the implementation of the project.

**Author Contributions:** B.F. conceived and designed the whole structure of the paper and wrote the paper; Y.M. accomplished the data collation. All authors have read and agreed to the published version of the manuscript.

**Funding:** This research was funded by young teachers' scientific research ability improvement program of Northwest Normal University in 2018 (No. NWNU-LKQN-18-4).

**Institutional Review Board Statement:** Not applicable.

**Informed Consent Statement:** Not applicable.

**Data Availability Statement:** Not applicable.

**Conflicts of Interest:** The authors declare no conflict of interest.

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
