# Peer review of "Network Construction for Overall Protection and Utilization of Cultural Heritage Space in Dunhuang City, China"

_sustainability, doi:10.3390/su15054579_

Round 1

Reviewer 1 Report

This paper examined Dunhuang as a case study and assessed the adequacy of its constituent factors' conservation and exploitation. For this study, a two-way index of conservation and utilization was constructed. First, I suggest the authors have the article edited by a native speaker. There are several grammatical problems, for example, line 14, should be “protection, as well as…”

Punctuation also needs to be checked; for example, in line 59, in . should be placed before ”

And some direct quotes did not cite sources, for example, in line 64.

A statement such as “On this basis, we hold that cultural heritage space as a whole is made up of the 67 heritage itself as well as its overall environment” is very obvious and unoriginal.

Reviewer 2 Report

 Network Construction for Overall Protection and Utilization of Cultural Heritage Space in Dunhuang City, China

 Bin Feng  and Yongchi Ma

The paper deal with the evaluation of the suitability of conservation factors and utilization factors through a two-way index of conservation and utilization were evaluated. In addition, the suitability characteristics of conservation and utilization using a field strength model that considered various factors in local cities in China were analyses.

Abstract

Please structure the abstract as:

Introduction-Aims

Method

Results and interpretation

Introduction:

Literature review is missing.

Please develop the literature review and update to 2022

Please analyses critically the findings of the articles and the limitations.

Please indicate also at least tree similar article to your research published recently (last 5 years).

Aim of the study is not clear indicated

Fig 1 line 170: please revise the legend. The characters of the written is different. The colors are not visible enough, please revise.

Table 1 line 206: please insert the datasource.

Eq1- lines 215-216, please insert the datasource and the authors.

Eq 2,3,4  idem

Figs 2 and 3 lines 313. Please insert the datasource (authors of the photos, date you took the photos)

Figs 4, 5, 6 please insert the datasources. Authors elaboration?

Methodology

Flowchart of the methodology steps to be inserted

Please explain better why you took into consideration the  Index System for the Suitability Evaluation.

What is the main question addressed by the research?

The main question addressed. The subject to which the paper address is very actual one, Post pandemic period. The paper review of wide range of articles (but must develop and internationalize  the list) in this research domain; their advantages and disadvantages etc are not enough exploited in a special subchapter of literature review.

Is it relevant and
interesting?

The paper is relevant especially nowadays in Post-pandemic in the complex environment of Dunhuang, China, concerning the Network Construction for Overall Protection and Utilization of  Cultural Heritage Space

How original is the topic?

Is an actual topic with medium degree of originality; but it is important subject especially in nowadays period, authors findings can contribute successfully to management models.

What does it add to the subject
area compared with other published material?

The paper is not very  well documented because the authors cited 37 scientific published articles; we suggested to develop and to be updated to 2022.

We consider useful for the paper also the following published article. Please see and cite it:

Ilieş Marin, Ilieş Dorina, Josan Ioana, Ilieş Alexandru, Ilieş Gabriela, (2010), The Gateway of Maramureş Land. Geostrategical Implications in Space and Time, in Annales. Annals for Istrian and Mediteranian Studies, Series Historia et Sociologia, ISSN 1408-5348, 20, 2010, 2, Zalozba Annales, Koper, Slovenia, pag. 469-480, (http://www.culture.si/en/Annales_Journal)

Ilieş Alexandru, Grama Vasile, (2010), The external western Balkan border of the European Union and its borderland: Premises for building functional transborder territorial systems, in Annales. Annals for Istrian and Mediteranian Studies, Series Historia et Sociologia, ISSN 1408-5348, 20, 2010, 2, Zalozba Annales, Koper, Slovenia, pag. 457-469, (http://www.culture.si/en/Annales_Journal)

Is the paper well written?

The paper is well written. The quality of English translation is good.
Is the text clear and easy to read?

The text is not very well structured. Methodology and discussions sections must be seriously revised.

Are the conclusions consistent with the evidence and arguments presented?

The results and discussions subchapters must be developed.

Best regards,

December 2022

Reviewer 3 Report

The article presents a detailed and precise study on the state of conservation and use of heritage in Dunhuang city, Gansu Province, China. The purpose of the author and his collaborator is to present a model of the network for overall protection and utilization taking into account a wide range of factors in the equation. The narrative style is direct and concise. In general, it is a good article that deserves to be published in this journal. Before its publication, some proposals are recommended that could contribute to improve its academic impact.

- The main argument is the lack of clear correspondence between the methodologies section and the results section, in relation to the high number of factors that are introduced into the equation and the scant "product" obtained from these mathematical operations at point 3. A table with explanatory values ​​could be added and include references to the data obtained.

- The paragraph (442-448) in which Chinese characters are introduced could also be modified, since unfortunately those of us who do not know this language do not know what it refers to.

It would also be convenient to increase the international bibliographical references.

In addition, in line 417 there is a "1" left over.

Reviewer 4 Report

This paper aims at effective protection and rational utilization of cultural heritage. A case study has been presented and analysed. The paper provides some useful findings and observations, although some comments should be addressed as listed below.

1. How do the authors ensure that the adopted evaluation model presented in Section 2.3 is reliable for this study?  

2. Fig. 2: Is it true that every conservation index is related to factors of cultural heritage to utilization index? It looks quite messy and unclear.

3. The latest research studies must be included in the literature review, for example:

Yu, S., Zhang, Q., Hao, J. L., Ma, W., Sun, Y., Wang, X., & Song, Y. (2023). Development of an extended STIRPAT model to assess the driving factors of household carbon dioxide emissions in China. Journal of Environmental Management, 325, 116502.

4. A table may be needed to provide quantitative assessment to complement the evaluation process presented in Fig. 5.

Round 2

Reviewer 1 Report

NA

Round 3

Reviewer 1 Report

na